# Agro-Waste Bean Fibers as Reinforce Materials for Polycaprolactone Composites

Cristina De Monte [1], Leonardo Arrighetti [1], Lucia Ricci [1,*], Alessandra Civello [1,2] and Simona Bronco [1]

1  Istituto per i Processi Chimico-Fisici, Sede di Pisa del Consiglio Nazionale delle Ricerche (CNR-IPCF), Via G. Moruzzi 1, 56124 Pisa, Italy; cristina.demonte@pi.ipcf.cnr.it (C.D.M.); leonardo.arrighetti@pi.ipcf.cnr.it (L.A.); simona.bronco@pi.ipcf.cnr.it (S.B.)
2  Dipartimento di Scienze Agrarie, Alimentari e Agro-Ambientali, Università di Pisa, Via del Borghetto 80, 56124 Pisa, Italy
*  Correspondence: lucia.ricci@pi.ipcf.cnr.it

**Abstract:** The agrifood industry shows one of the widest ranges of possible end products from crops, such as fruits, legumes, cereals, and tubers. The raw material is generally collected and processed industrially, producing a significant amount of organic waste. The overall picture is made more complex by the wide variety of nature and composition, and by the difficulty identifying the possible uses of the wastes coming from the processing industry. Such wastes are often disposed of in landfills or treated in waste-to-energy plants depending on the area where they are produced. The circular economy approach has suggested numerous possible generic strategies to improve waste management, involving the exploitation of waste to obtain new value-added products. The use of fibers from legume waste from the canning industry in the bioplastics production sector is a promising and relatively little explored line, particularly for the fibers of beans and green beans. With this in mind, in this article, green bean and borlotti bean fibers obtained from the treatment of wastes were used as reinforcing material for polycaprolactone (PCL)-based biocomposites by melt blending. Analyses were carried out about the morphological, spectroscopic, thermal, and mechanical properties of the starting and the obtained materials.

**Keywords:** bean fibers; polycaprolactone; biocomposites





## 1. Introduction

The unsustainable use of resources results in their incipient depletion. The concern about resource depletion is exacerbated by the rapid growth of the population, which in 2021 reached 7.9 billion people, and currently, it is estimated to exceed 8 billion [1]. Accordingly, global food waste production steeply increased, and in this situation, a more aware use of residues, waste, and byproducts of the processes exploiting primary resources can be a possible relief to depletion (Figure 1). Actually, in the last decade, the valorization of agrifood wastes attracted attention for two main reasons: (i) achieving better waste management with obvious economic savings on disposal costs and (ii) economic returns from products with high added value, such as cosmetics, pharmaceutics, chemicals, animal feed, and additives. In agrifood industries, one of the higher worldwide productions is that of legumes; in particular, worldwide dry beans production amounted to about 28 Mtons (0.4 Mtons from Europe) in 2021, while that of green beans amounted to about 23.4 Mtons (1.2 Mtons from Europe) [2].

The development of innovative plastic composite using agro-waste coming from the latter production chain, giving rise to biodegradable plastics such as Polylactic acid (PLA), Polycaprolactone (PCL), or Polyhydroxyalkanoate (PHA), has gained importance over the years thanks to the possibility to reduce production costs and to improve the sustainability of processes using less costly materials, achieving the required properties integrating agrifood-wastes and byproducts. The selection of biodegradable plastic as

a polymer matrix in the composites complies with the best management of end-of-life products in order to obtain a minor environmental impact than that of petroleum-based and non-biodegradable plastics. According to European Bioplastics, the production of bioplastics in Europe in 2022 amounts to about 2.22 Mtons, while a sharp increase of production was estimated up to 6.30 Mtons in 2027, with an increase in biodegradable fraction from 51.5% to 56.5% [3]. Among the possible green polymers, PCL is one of the most studied and promising polymers in many fields. It is successfully used in biomedical applications and packaging such as compost bags. PCL is an aliphatic polyester produced by ring-opening polymerization (ROP) of ε-caprolactone. One of the main problems of PCL concerns its cost of production, but this limitation can be overcome by including the processing of compounding with cheaper materials such as agrifood waste. PCL has a low glass transition temperature ($-60\ ^{\circ}$C) and a low melting point between 59 and 64 $^{\circ}$C. These low temperatures, required for melt blending processing, constitute an excellent synergy with the use of natural fibers [4–6]. At these temperatures, the fibers retain their properties better and tend not to darken and stiffen. As for natural fibers, they are mainly obtained from plants (such as sisal, flax, hemp, jute, and cotton), minerals, or animals (e.g., wool). The fibers of vegetable origin, in particular, have a highly variable composition depending on the nature of the plant species and the season, the strong influences of which affect both the chemistry and the quality of the supply. The main components of vegetable fibers, even with variable ratios, are waxes, cellulose, hemicellulose, pectin, and lignin [7]. This latter has a radically different nature with respect to the others. In fact, it is mainly formed by phenols with different crosslinking grades and contributes to the recalcitrance of biomass to processing treatments. Considering the general preparation of composites, one of the key factors that mainly affects the final properties is the interaction between additive and polymer matrix on interfaces. In particular, depending on the hydrophobicity of the polymer matrix, the interactions with fibers (such as cellulose, which is hydrophilic) may be poor, resulting in bad dispersion, inhomogeneity, and defects, which may adversely affect the mechanical and structural proprieties of the materials [8]. These limits could be potentially overcome using fiber pretreatments of biomass. Several articles deal with various types of fiber pretreatments with alkali (sodium hydroxide) [9] or oxidative agents (potassium permanganate, peroxide) as well as with reagents such as stearic acid [10], which enhance the properties, making the fiber surfaces more compatible with the polymer matrix [11] reacting with hydroxyl groups of cellulose/hemicellulose.

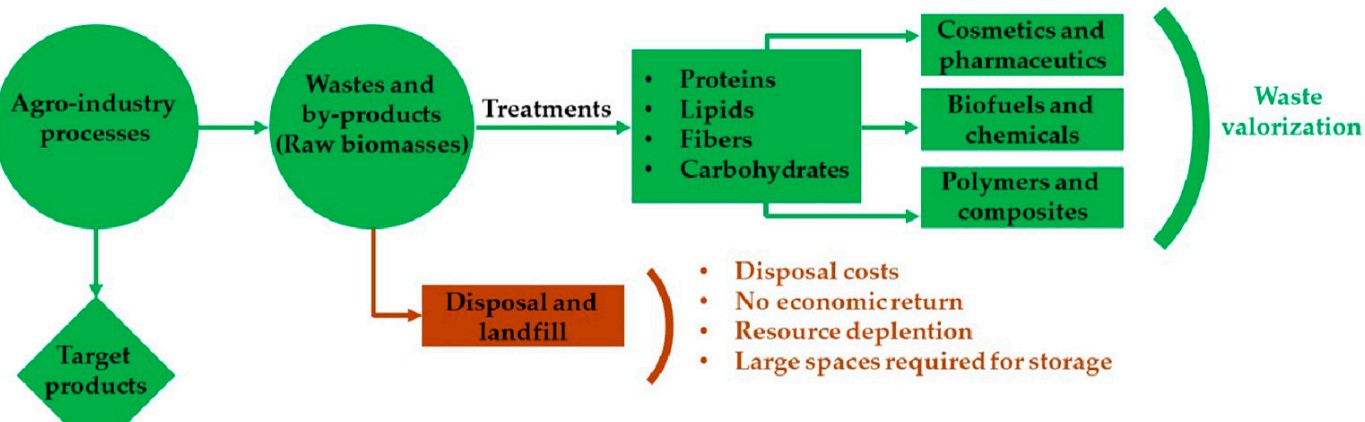

**Figure 1.** The schematization of possible effects of waste valorization on a general process.

In this work, green bean and borlotti bean fibers obtained from legume waste treatment [12] were used as reinforcing material for the preparation of Polycaprolactone (PCL)-based biocomposites by melt blending. Fiber fractionation was carried out in order to estimate fibers' average chemical composition to rationalize the possible effects found on

composites produced by using raw fibers. Morphological, spectroscopic, thermal, and mechanical analyses were carried out on the starting and final materials.

Although there are indications that it is possible to use fibers from peas, beans, and chickpeas [13], there is no evidence in the literature on the real use of bean fibers for the preparation of biocomposites, except for soybeans [14–16]. Cinelli et al. [17] used post-consumer peas fibers for PHA. Recently Novais Miranda et al. [18] added 0, 10, 20, 30, and 40% bean residual agricultural waste to recycled low-density polyethylene (LDPE). The bean residues used in Novais Miranda's experiment consisted of roots, stems, and pods that were collected after natural drying. To produce the composites, the residue was ground and separated according to granulometry. The selected particles passed through a 40-mesh sieve and retained on a 60-mesh sieve.

## 2. Materials and Methods

### 2.1. Materials

Two types of fiber, obtained from byproducts of borlotti beans (named B1 in the following) and green beans (named G1 in the following) supplied from Stazione Sperimentale per l'Industria delle Conserve Alimentari—SSICA Parma, Italy, were used as received. Polycaprolactone (PCL) Capa™ 6500 was supplied from Perstorp Holding AB, Malmö, Sweden (molecular weight is approximately 50,000 g mol$^{-1}$, supplied in granular form-approx. 3 mm pellets, MFI 7 g/10 min. with 2.16 kg, water content < 1.0%). Capa™ 6500 is used in a variety of adhesive applications and is compatible with a wide range of common thermoplastics and soluble in several common solvents.

Toluene RPE grade, Ethanol (96% grade), Acetic Acid glacial, ACS reagent, ≥99.7%, Sodium Acetate ACS reagent, ≥99.0%, Sodium metabisulfite ReagentPlus®, ≥99%, Sodium hypochlorite solution and Potassium hydroxide ACS reagent, ≥85%, pellets were purchased from Sigma Aldrich, Milan, Italy. Sodium carbonate was supplied from Carlo Erba, Milan, Italy and di-Ammonium oxalate monohydrate 99.0–101.0% crystals were supplied from J. T. Backer, Milan, Italy. Ultrapure Water was obtained from Merck Millipore productor, Darmstadt, Germany.

All reagents and solvents were used as received.

### 2.2. Fiber Fractionation

Fiber samples were treated to separate and quantify the fractions of wax, polysaccharide, and lignin [19]. All the treatment steps are outlined in Scheme 1. Firstly, about 5 g of each type of fiber were washed using 250 mL of washing solution (1.25 g of commercial soap and 1.25 g of Na$_2$CO$_3$) at 70 °C under stirring for 1 h as pretreatment in order to remove impurities on the surface and inside the mass of fibers (Step 1). Afterward, fibers were washed with distilled water, filtered, and then dried under vacuum conditions for 3 h and then kept in an oven (Steril Stiv, COBAMS SRL, San Lazzaro di Savena (BO), Italy) at 60 °C for 12 h (borlotti beans, B1-soap) or lyophilized (green beans, G1-soap) to constant weight (freeze dryer Thermo Fisher Scientific Inc. HETO POWERDRY LL 1500,Waltham, MA, USA). After the pretreatment, the extraction process consisted of 4 different and progressive steps:

- Wax extraction;
- Pectin extraction;
- Lignin extraction;
- Cellulose and Hemicellulose extraction.

Wax quantification (Step 2)

About 4 g of pretreated cleaned fibers (B1-soap and G1-soap) were treated with 400 mL of toluene/ethanol mixture (2:1 *v/v*) under stirring at room temperature all night long. Afterward, fibers were washed with 100 mL of the same mixture and with Ethanol at room temperature. Fibers were then filtered and dried at 100 °C (borlotti beans, B1-Tol-EtOH) or lyophilized (green beans, G1-Tol-EtOH) until constant weight. The wax

fraction was estimated by Equation (1) from the percentage weight variation after the extraction procedure:

$$W\% = \frac{m_0 - m_1}{m_1} * 100 \tag{1}$$

where $m_0$ is the weight of the pretreated cleaned fibers (g), and $m_1$ is the weight of fibers after the first extraction step (g).

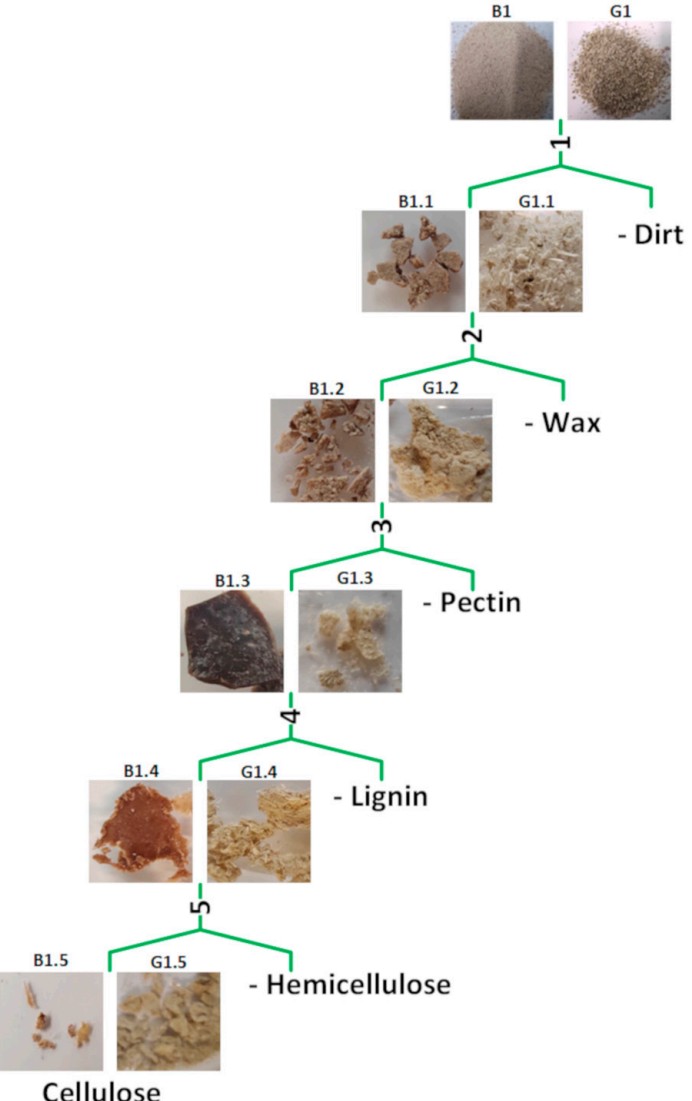

**Scheme 1.** Fiber fractionation.

Pectin quantification (Step 3)

B1-Tol-EtOH and G1-Tol-EtOH were further processed in order to quantify pectin in their bulk structure. About 3.5 g of each collected fiber was suspended in 350 mL of ammonium oxalate $(NH_4)_2C_2O_4$ solution (0.5% wt). The treatment was conducted at 80 °C under stirring for 72 h. Samples were then washed with distilled water at 50 °C and dried under vacuum at 40 °C (borlotti beans, B1-Oxa) or lyophilized (green beans, G1-Oxa). Pectin fraction was estimated by Equation (2) from the weight variation after the extraction procedure:

$$P\% = \frac{m_1 - m_2}{m_2} * (100 - W\%) \tag{2}$$

where $m_1$ is the weight of fibers after the first extraction step and $m_2$ is the weight of fibers after the second extraction step.

Lignin quantification (Step 4)

About 2 g of B1-Oxa and G1-Oxa was treated with 100 mL of a sodium hypochlorite NaClO 0.7% wt solution (pH 4, acetic buffer). Afterward, the obtained suspension was heated at 90 °C for 1.25 h, and then 25 mL of sodium bisulfite $Na_2S_2O_5$ solution at 2% wt solution was added, and the mixture was stirred for another 15 min. Treated fibers (B1-Buffer and G1-Buffer) were washed with distilled water and dried at a constant weight under vacuum conditions (0.01 mmHg).

The percentage amount of lignin fraction was estimated by Equation (3):

$$L\% = \frac{m_2 - m_3}{m_3} * (100 - W\% - P\%) \tag{3}$$

where $m_2$ is the weight of fibers after the second extraction step in both cases, and $m_3$ is the weight of fibers after the third step.

Cellulose and hemicellulose quantification (Step 5)

About 1.5 g of B1-Buffer and G1-Buffer were treated with KOH solution (24%) (fiber ratio of 1% wt) for 4 h in an ice bath. The solid residue (B1-KOH and G1-KOH) was filtered and washed at first with 2% acetic acid solution, then with distilled water, and finally dried up to constant weight. The solution was then drawn under vacuum with a Rotavapor and mechanical pump until constant weight, and calculated hemicellulose fraction percentage amount was estimated by Equation (4):

$$H\% = \frac{m_3 - m_4}{m_4} * (100 - W\% - P\% - L\%) \tag{4}$$

where $m_3$ is the weight of fibers after the third steps evaluated, and $m_4$ is the weight of fibers after the fourth step.

The cellulose fraction was finally calculated by Equation (5):

$$C\% = 100 - W\% - P\% - L\% - H\% \tag{5}$$

## 2.3. PCL Composite Preparation

PCL pellets and fibers were pre-dried at 50 °C overnight (over 12 h–720 min) in vacuum condition to remove humidity absorbed (vacuum oven 1GV High-performance Vacuum Ovens, Fratelli Galli, Fizzonasco di Pieve Emanuele (MI), Italy). Under this condition, Mella et al. [20] obtained residual moisture in beets of less than 0.1% after 500 min. Beatrice et al. [21] dried PCL at 40 °C under vacuum for 4 h before being processed in a mixer.

Composites were prepared using the discontinuous mixer Brabender Plastograph EC (Brabender GmbH & Co. KG, Duisburg, Germany), with a mixing chamber of 30 cm³, interfaced with Brabender Mixing Software "Win Mix 1.0" for data management and acquisition. Two composites were prepared for each fiber type from borlotti beans (B1) and green beans (G1) with 5% wt and 40% wt fiber content, respectively. To evaluate the possible effects of different amounts of fibers added, all the composites were processed at 50 rpm for 10 min at 90 °C. The addition of the fibers to the mixing chamber was made two minutes after the addition of the PCL pellets in order to ensure the complete melting of the polymer material during the processing. Sample films were prepared by compression molding using a Carver No3851CE hot melt hydraulic press (Carver Inc., Wabash, IN, USA) at 120 °C from the granules collected at the end of the mixing procedure. Two folded sheets of Teflon were placed between the two heated plates to prevent their damage. About 1 g of each sample was melted for 2.5 min without applied pressure and then compressed at 2 tons for 1.5 min. The cooling of the films after pressing is carried out, keeping the material in the air.

### 2.4. Characterization Methods

Each sample was analyzed by Infrared Spectroscopy by using a Jasco FT/IR6200 spectrometer (Jasco, Tokyo, Japan) equipped with ATR accessory PIKE MIRacle (Madison, WT, USA). A total of 64 scans from 4000 to 650 cm$^{-1}$ were collected for each sample.

Scanning Electron Microscope (SEM) images were recorded with JEOL JSM-5600LV (Jeol, Tokyo, Japan) microscope operating at the Department of Civil and Industrial Engineering, University of Pisa. Specimens were prepared by cryogenic fracture performed by immersion of a piece of each film in liquid nitrogen for 4 min and quickly cut by cooled scissors.

Differential Scanning Calorimetry (DSC) analyses were performed with an SII DSC7020 EXSTAR Seiko calorimeter (Seiko, Chiba, Japan) under nitrogen inert atmosphere fluxed at 100 mL/min during all measurements on the raw fiber and their treated fractions in open pans according to the following thermal protocol: 1° heating step from 20 to 120 °C at 10 °C/min (hold 2 min); 1° cooling step from 120 °C to 0 °C at 10 °C/min (hold 2 min); 2° heating step from 0 °C to 400 °C at 10 °C/min (hold 2 min); second cooling step from 400 °C to 20 °C at 30 °C/min (not registered). The analyses of PCL composites were carried out according to the following thermal protocol: 1° heating step from 20 to 100 °C at 10 °C/min (hold 5 min); 1° cooling step from 100 °C to −100 °C at 10 °C/min (hold 2 min); 2° heating step from −100 °C to 100 °C at 10 °C/min (hold 2 min); second cooling step from 100 °C to 20 °C at 30 °C/min (not registered). The sample amount used for all DSC was about 5 mg. The degree of crystallinity ($\chi$%) of polymer composites was estimated by using Equation (6):

$$\chi\% = \frac{|\Delta H_m|}{\Delta H_m^\circ} * \frac{100}{x} \tag{6}$$

where $x$ is the weight percentage amount of PCL in polymer blends, $\Delta H_m$ is the measured melting enthalpy of samples, and $\Delta H_m^\circ$ is the theoretical melting enthalpy value of 100% crystalline PCL (135 J/g [22]).

Thermogravimetric analysis (TGA) was performed by an SII TG/DTA 7200 EXSTAR Seiko analyzer (Seiko, Chiba, Japan) under nitrogen inert atmosphere fluxed at 200 mL/min during all measurements from 30 to 700 °C at a 10 °C/min rate on about 5 mg of materials in alumina pans.

Tinius Olsen H10KT dynamometer (Horsham, PA, USA) equipped with a 500 N load cell QMAT was used for mechanical stress–strain characterization. Sample specimens were obtained by die-cutting the composite's films according to ASTM D638. Films were then conditioned at rh 50% and 23 °C for 48 h. The traction speed was set to 10 mm/min. Data sets of the processing parameters study were screened by one-way ANOVA, and a Tukey test was used for post hoc analysis; significance was defined as $p < 0.05$.

### 3. Results and Discussion

#### 3.1. Analyses of Fibers: Chemical Fractionation and Characterizations

In spite of the fact that green beans (G1) and borlotti beans (B1) fibers are different varieties of the same species, i.e., *Phaseolus vulgaris* L., the morphology of their fibers showed clear differences, as shown by the SEM images reported in Figure 2. In green beans (Figure 2, left), the defined longitudinal form of fibers is shown, while for borlotti beans, the form appears more spherical. The adhesion and agglomeration between green bean fibers, densely packed in bundles of cellulose fibrils, were appreciable. This morphology, in general, promotes the interaction between fibers and the other structural components of the plants. Round and elliptical granules, with dimensions of 25–50 μm, were observed and measured in borlotti fibers (Figure 2, right). These granules are probably formed by starch, and their forms are typical of *Phaseolus vulgaris* species [23]. It is also possible to find the presence of crackles on the surface of the granules, which is probably due to extraction processes.

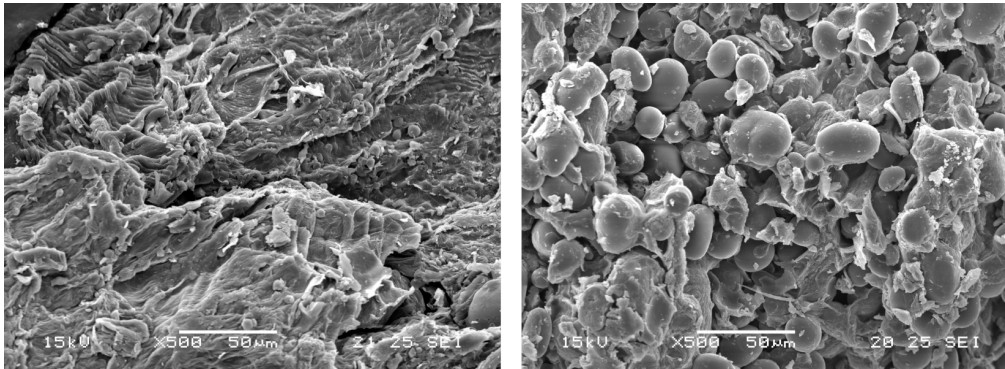

**Figure 2.** SEM micrograph of green bean fibers (G1, **left**) and borlotti bean fibers (B1, **right**).

Both types of fiber, from borlotti beans and green beans, respectively, were treated independently under the same extraction protocol [11]. Absorbed water was removed through freeze-drying because of the greater hygroscopicity of green bean fibers. Drying in the oven was considered enough for borlotti beans.

Important differences emerged (Table 1) by comparing the composition of the two types of fibers. Borlotti fibers are formed mainly from hemicellulose (68%), while cellulose is the major component of green bean fibers (52%). The different hemicellulose/cellulose ratios may radically imply different effects on the mechanical properties of a generic composite containing these fibers. Hemicellulose acts as a glue between cellulose and lignin, while cellulose acts as a sustaining structure, and lignin, with its phenolic structure, gives the material recalcitrance to extractive agents. The different content of pectin can affect the properties of the composites. In general, pectin is the most hydrophilic component of vegetal fibers, and its gelatinous nature gives flexibility to the structure of the plant. Green bean fibers result in being richer in pectin content than borlotti bean fibers (27% vs. 11%, respectively), coherently with the major water uptake of the green bean fibers. These low composition differences may imply important differences in the potential effect of the use of this fiber in composites.

**Table 1.** Percentage composition of the fiber components of borlotti beans and green beans.

| Component | Borlotti Beans (% wt) * | Green Beans (% wt) * |
|---|---|---|
| Waxes | 5 | 8 |
| Pectin | 11 | 27 |
| Lignin | 9 | 6 |
| Hemicellulose | 68 | 7 |
| Cellulose | 7 | 52 |

* Values rounded to unity.

All the fractions were then characterized by spectroscopic, thermal, and morphological analyses, with the exception of the B1-KOH fraction, which resulted in too much adherent to the filter paper.

The chemical nature of the surface of B1 and G1 fibers was analyzed by ATR spectroscopy. The spectra of the raw fibers (thick black lines) are compared in Figure 3, collecting spectra after each step of fractionation. The typical bands of the main components, such as cellulose, hemicellulose, pectin, and lignin, of common vegetal fiber were recognized. The spectrum of G1 (Figure 3, left) showed a vibration mode assigned to –OH stretching of cellulose ($\alpha$-form) at 3744 cm$^{-1}$. The large absorption band centered at 3346 cm$^{-1}$, common to all the collected spectra, is attributable to –OH stretching mode in interchain interactions. The C–H stretching mode at 2930 cm$^{-1}$ is characteristic of cellulose and hemicellulose. Also, the absorption band centered at 2889 cm$^{-1}$ is characteristic of hemicellulose vibrational modes. The band due to the C=O stretching mode at 1740 cm$^{-1}$ is attributed to the

presence of –COOH of lignin and aldehyde in hemicellulose. After the lignin extraction (Step 3), a decrease in its absorption intensity was found. The absorption band centered at 1640 cm$^{-1}$ is attributed to the partial hydrophilic nature of fibers; in fact, its nature may be attributed to the presence of absorbed water which is progressively removed during the extraction steps. A contribution to the intensity of the latter signal may be due to the –NH bending of protein residues present at the surface of the fibers that remain after the component's extraction from the raw legumes. Multiple smooth absorption bands were observed in the fingerprint region between 1400 and 1300 cm$^{-1}$, whose presence may be associated with the C–O groups of the guaiacyl ring breathing in polysaccharides and C–O stretching vibration of the acetyl group in lignin. The sharp band at 1021 cm$^{-1}$ is assigned to ethers C–O stretching of cellulose [24]. ATR analysis of borlotti bean fibers (Figure 3, right) reveals similar bands. In addition, absorption bands are observed at 3340 cm$^{-1}$, 2925 cm$^{-1}$, 1744 cm$^{-1}$, 1644 cm$^{-1}$, and 1210 cm$^{-1}$. These bands are similarly attributed to OH stretching due to interchain interactions or –NH of protein residues, C=O stretching of lignin and hemicellulose, absorbed water or protein residue, and C–O stretching, respectively. An important decrease in the intensity of the absorption of the bands at 1744 cm$^{-1}$ and 1644 cm$^{-1}$ was observed after the third extraction step in B1 (B1-Oxa).

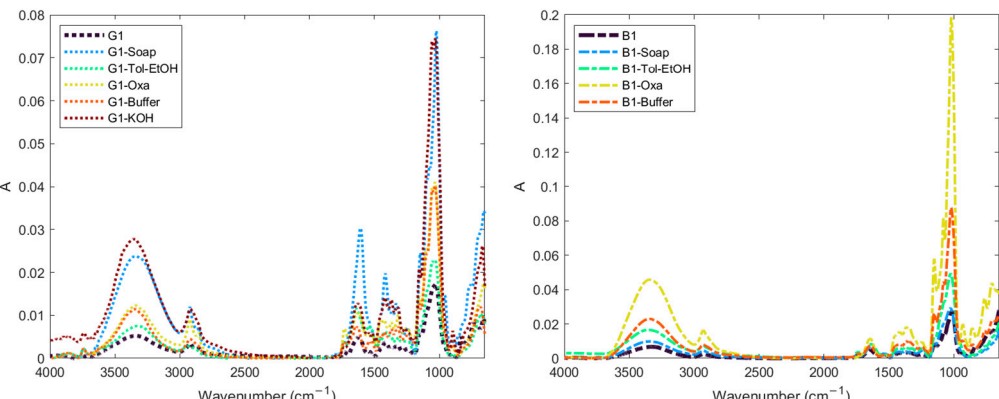

**Figure 3.** Overlapping of ATR-FTIR spectra of green (**left**) and borlotti (**right**) bean fibers after each treatment of fractioning.

Thermal analyses DSC showed for both fibers a peak at 120 °C on the first heating (Figure 4—thin line) attributable to the evaporation of the absorbed water. As the extraction steps on G1 proceed, this peak progressively shifted to a lower temperature, probably due to the progressive removal of hydrophilic fractions from the bulk of the fibers and the depletion of water interactions which causes the water desorption to become easier. The same effect was observed on B1. The analysis of the second heating (Figure 4—thick line), instead scanned up to 400 °C, showed the shift to lower temperatures from 100 °C to 150 °C. According to the literature [25], the thermal transition at 270 °C is attributable to pectin degradation, also considering that after the oxalate treatment, the peak was no longer observed. The transition above 300 °C can be instead assigned to cellulose [26].

In order to evaluate the effect of the single extraction steps on the thermal stability of the fibers, thermogravimetric analysis was performed (Figures 5 and 6 and Tables 2 and 3). In particular, it investigated the different thermogram trends as a function of the progressive removal of the different fractions. Generally, natural fibers undergo thermal degradation in three or four steps on the basis of their chemical composition [26]. The weight loss stages as a function of the increasing temperature were due to: (i) the evaporation of moisture and the absorbed water content, (ii) the degradation of hemicellulose, (iii) the degradation of cellulose, and finally (iv) the degradation of lignin. These processes occur between 100 °C and 500 °C and depend on the relative ponderal amounts of the single biomolecules in the bulk of the fiber.

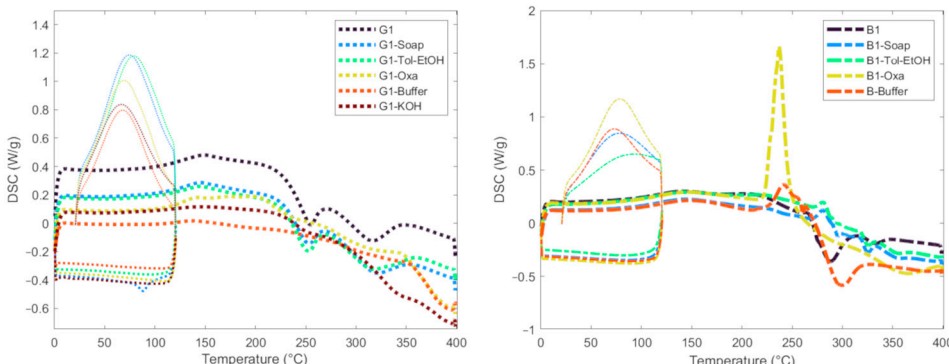

**Figure 4.** Overlapping of DSC thermograms of green (**left**) and borlotti bean fibers (**right**) after each treatment of fractioning.

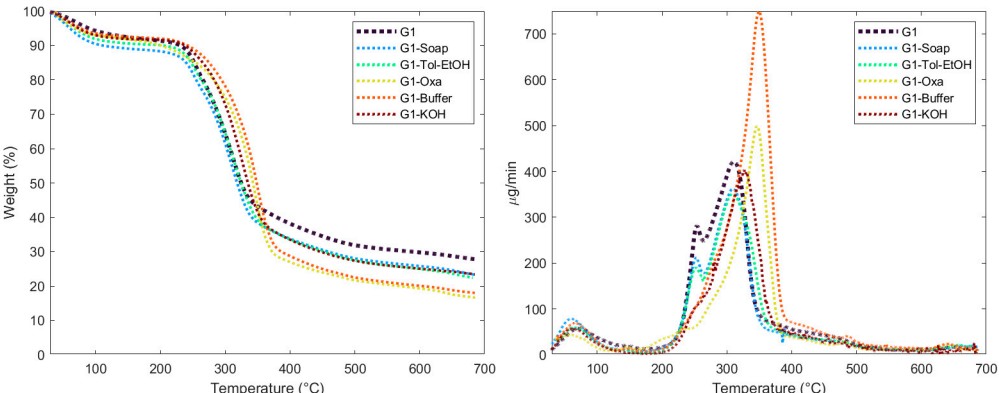

**Figure 5.** TGA thermograms (**left**) and the relative DTG (**right**) of the green bean fibers after each treatment of fractioning.

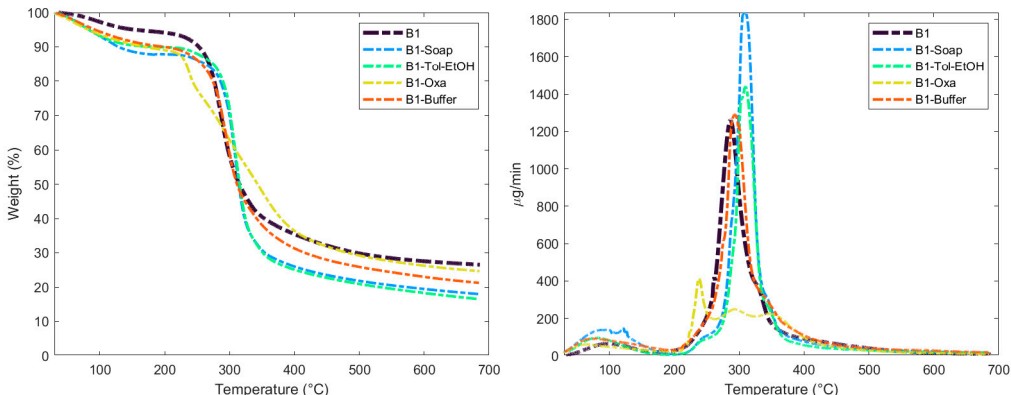

**Figure 6.** TGA thermograms (**left**) and the relative DTG (**right**) of the borlotti bean fibers after each treatment of fractioning.

**Table 2.** TGA results of the green bean fibers after each treatment.

| Samples | From TGA | | | From DTG | |
|---|---|---|---|---|---|
| | Water Loss % | $T_{onset}$ (°C) | Residue % | $T_{max1}$ (°C) | $T_{max2}$ (°C) |
| G1 | 8.0 | 235.4 | 27.9 | 252.8 | 310.1 |
| G1-Soap | 10.4 | 233.0 | 23.5 | 252.3 | 308.8 |
| G1-Tol-EtOH | 8.0 | 232.6 | 22.6 | 252.1 | 309.4 |
| G1-Oxa | 7.4 | 284.2 | 16.7 | - | 346.2 |
| G1-Buffer | 7.0 | 284.0 | 18.1 | - | 349.3 |
| G1-KOH | 8.0 | 266.9 | 23.6 | - | 326.7 |

**Table 3.** TGA results of the borlotti bean fibers after each treatment.

| | From TGA | | | From DTG | | | |
|---|---|---|---|---|---|---|---|
| Samples | Water Loss % | $T_{onset}$ (°C) | Residue % | $T_{max1}$ (°C) | $T_{max2}$ (°C) | $T_{max3}$ (°C) | $T_{max4}$ (°C) |
| B1 | 5.2 | 267.3 | 26.6 | 261.2 | 286.4 | - | 325.7 |
| B1-Soap | 12.0 | 276.2 | 17.9 | 244.3 | - | 308.2 | 341.2 |
| B1-Tol-EtOH | 9.7 | 280.6 | 16.5 | 243.1 | 303.4 | 309.6 | 340.8 |
| B1-Oxa | 10.7 | 226.0 | 24.7 | 226.0 | 293.2 | - | 345.9 |
| B1-Buffer | 9.7 | 273.1 | 21.3 | 277.8 | 292.9 | - | - |

For B1 fibers, the weight loss between 33 °C and 150 °C (Figure 6, left) was due to the loss of water, which can be adsorbed both on the surface of the fibers by capillarity and/or by the chemical interaction. For G1 fibers (Figure 5, left), no sharp trend was observed, and the water content remains essentially constant. As far as B1 fibers are concerned, water absorption seems to increase significantly from the first treatment, due to multiple exposure to the aqueous solution used in the chemical mixture of fractioning. Differences emerged in the comparison of the $T_{onset}$ trend as the fractioning proceeds. In general, it is well known that hemicellulose degradation is located between 200 to 350 °C while that of cellulose takes place between 270 and 350 °C. Lignin, in general, degrades between 300 and 430 °C due to its aromatic nature. These ranges are moreover affected by numerous factors, such as molecular weight of the biomolecules, inter/intra-chains interactions, and chemical composition of fibers. In this work, it was observed that $T_{onset}$ remained constant until the fourth treatment step (Oxalate treatment), in correspondence that its value raises to 284.2 °C. This change is coherent with the removal of the pectin fraction from fiber bulk which is known to be less thermally stable than the other fractions [25]. A peak at about 252 °C, which is more easily observed in the first derivative of absorption (Figure 6, right), disappears after the oxalate extraction step. The thermogram of the sample treated with KOH shows a single maximum on its first derivative at temperatures higher than 300 °C, which is attributable to the presence of cellulose as a major component. The main transition temperatures are collected in Tables 2 and 3.

At the end of the fractioning protocol, an increase in the thermal stability was therefore observed coherently with the removal of thermolabile fractions such as pectin. In general, the TGA/DTG profiles of raw G1 and B1 fibers indicated their processability up to 200 °C as another kind of natural fiber [27].

Considering this information, the choice of the processing temperature of 90 °C in the molten state with PCL should not have a degrading effect on the fibers.

*3.2. Preparation of Composites*

Four different composites were prepared with PCL, respectively, at 5% and 40% by fiber weight with respect to the polymeric matrix by mixing in the molten state. These two percentages were chosen to evaluate as extremely different percentages of fibers can affect the chemical, physical, mechanical, and morphological properties of the composites.

In this condition (50 rpm and 90 °C), the degradation of the polymer matrix is negligible, and the torque value is in line with the values reported in the literature [21]. The analysis of the profile of the torque curves (Figure 7) with respect to the behavior of neat PCL establishes the stability of the polymeric matrix during the mixing process. The limited degradation in the case of the composites could be due to the overlapping of two different trends. The decreasing of the torque, due to the polymer matrix degradation, and the increasing of the torque, due to the change in viscosity, are both caused by the addition of the fibers. These phenomena seemed more marked in PCL40G1 (Figure 7, dark green line) than in PCL40B1 (Figure 7, brown line). This could agree with the fact that G1 composition is richer in cellulose while B1 contains more hemicellulose.

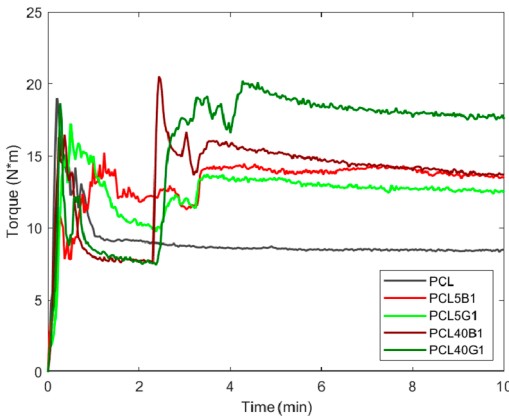

**Figure 7.** Torque curve of composites.

### 3.2.1. Morphological Properties of Composites

The observation of the cryogenic fractures captured by scanning electron microscope allowed us to qualitatively appreciate the arrangement taken by fibers in the bulk of the composite. Figure 8 shows, respectively, PCL5B1 and PCL40B1 fracture textures. Spherical/round agglomeration already observed for B1 fibers micrograph is maintained along the thickness of the fractures, and the extent of their presence was found to be consistent with the weight additions used. However, cavities formed by debonding and pulling out of fibers due to stress applied during the cryogenic fracture are present. Generally, this behavior indicates poor interaction between fillers and polymer matrix [28].

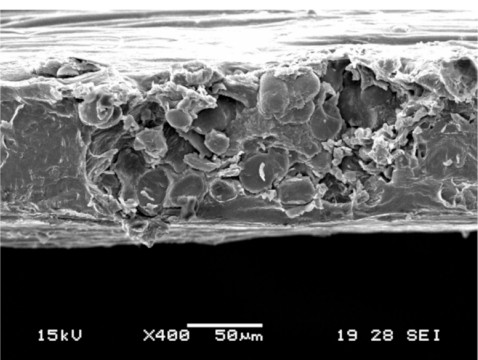 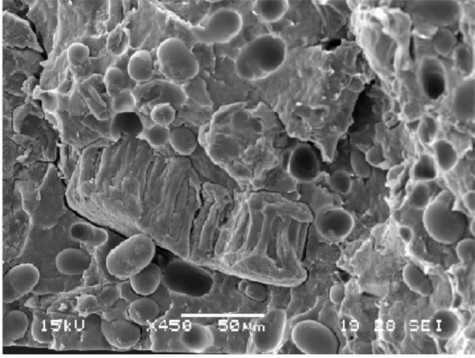

**Figure 8.** SEM images of PCL5B1 (**left**) and PCL40B1 (**right**).

A more oriented fibrous form of G1 is apparent in the micrographs collected for PCL5G1 and PCL40G1 instead (Figure 9, left and right, respectively). It seems that the polymer matrix wrapped the fibers during melt mixing processing; therefore, the system matrix fibers remain highly structured after solidification.

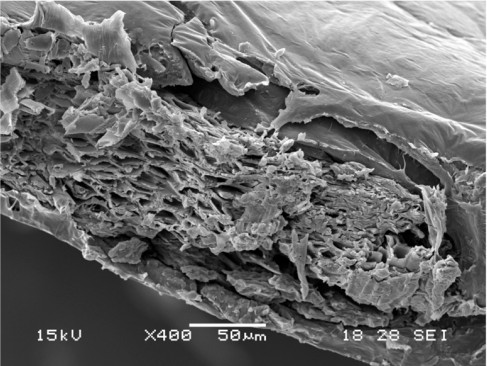 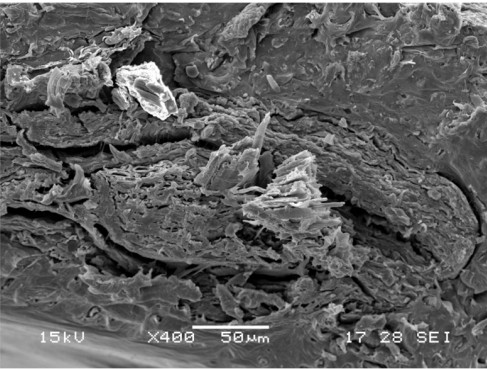

**Figure 9.** SEM images of PCL5G1 (**left**) and PCL40G1 (**right**).

### 3.2.2. Thermal Properties of Composites

The analysis of the DSC curve of neat PCL allowed us to appreciate the crystallization peak centered at 32.6 °C ($T_c$) on the first cooling curve of PCL (black line in the left image of Figure 10), while the glass transition temperature centered at −64.0 °C ($T_g$) and the melting peak centered at 57.4 °C ($T_m$) were calculated from the second heating curve (Table 4). These values are consistent with the literature data [29,30].

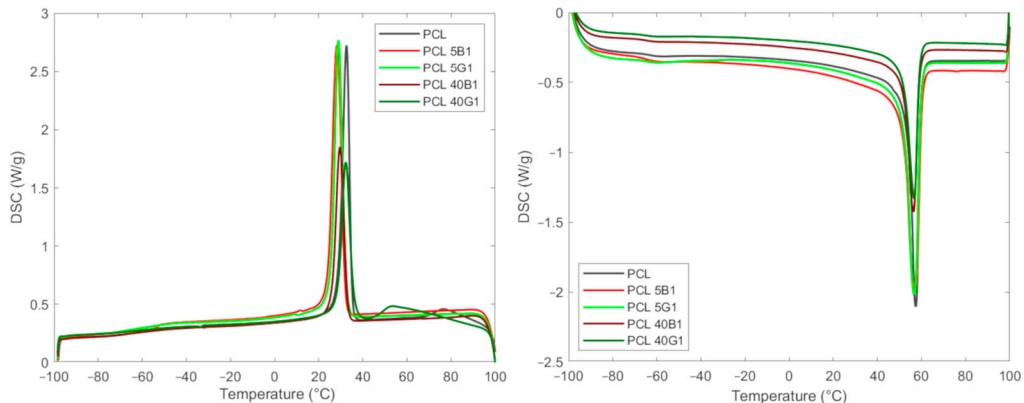

**Figure 10.** First cooling (**left**) and second heating (**right**) DSC curves of PCL reference and composites.

**Table 4.** Comparison of DSC results of PCL reference and composites.

| Samples | From 1° Cooling | | From 2° Heating | | | | |
| | $T_c$ (°C) | $\Delta H_c$ (J/g) | $T_g$ (°C) | $\Delta C_p$ (J/g °C) | $T_m$ (°C) | $\Delta H_m$ (J/g) | $\chi\%$ * |
|---|---|---|---|---|---|---|---|
| PCL | 32.6 | −63.4 | −64.0 | 0.085 | 57.4 | 66.7 | 49.4 |
| PCL5B1 | 28.4 | −78.3 | −64.7 | 0.14 | 56.9 | 92.2 | 71.9 |
| PCL40B1 | 29.4 | −48.9 | −64.5 | 0.10 | 56.3 | 58.0 | 71.6 |
| PCL5G1 | 29.0 | −76.5 | −64.6 | 0.11 | 56.6 | 84.3 | 65.7 |
| PCL40G1 | 32.3 | −44.7 | −65.0 | 0.12 | 56.7 | 51.5 | 63.6 |

* $\chi\%$ calculated with Equation (6) ($\Delta H_m^\circ$ = 135 J/g).

In the literature, it is extensively reported that the introduction of fibrous system on composites can affect crystallization dynamics in two different ways depending on fibers dimensions and polymer matrix:

1. Improving crystallinity by nucleating action [31–33];
2. Hindering crystallization by reducing chain mobility makes reorganization into ordered structures harder [34].

In this paper, we established that adding both G1 and B1 fibers up to a high content seems not to modify the values of $T_g$ (and relative $\Delta C_p$), $T_c$, and $T_m$ values, as extrapolated from DSC curves of composites (Table 4). On the other hand, changes were observed in crystallization enthalpy $\Delta H_c$ (and consequently on melting enthalpy variation $\Delta H_m$ and crystallization degree), independent of the nature of the fibers. Samples with 5% wt of fibers (PCL5B1, PCL5G1) showed a higher crystallization enthalpy (considering the absolute value) than that of neat PCL. The increase in the change of enthalpy associated with crystallization is correlated to a higher amount of released energy during the process (formation of an ordered structure from a disordered one, with subsequent decreasing of internal energy), which may be coherent with a possible nucleating action of the added fibers [35]. Considering the different ponderal amounts of polymer used in the composite's preparation at different fibers percentage, the crystallization degree seems to increase in all cases, especially for samples with B1-added fibers, indicating that fibers may act as enhancing support structures for polymer crystallization. A similar increase in $\chi\%$ was observed by del Ángel-Sánchez et al., which evaluated the effect of the addition of $TiO_2$ nanoparticles that purposely act as nucleating agents [36], even if the high effectiveness

of the nanoparticles is widely known to be due to their high surface even at very low concentration and petroleum-based polymer matrix [37].

The influence of the presence of natural fibers was also investigated by thermogravimetric analysis. PCL shows different mechanisms of thermal degradation depending on molecular weight and the process atmosphere. In particular, it was reported that in a nitrogen atmosphere and high molecular weight, thermal degradation takes place at about 390 °C through a single step with the formation of hexanoic acid and carbon dioxide (Figure 11, left) [38–40]. The value of $T_{onset}$ decreases towards values similar to those found for the respective used fibers. The presence of additional peaks in the composites with the highest percentage of fibers is apparent in the DTGs (Figure 11, right). Despite the shift to lower temperatures of $T_{onset}$ at all fibers percentage, the higher fiber content (40% wt) composites seem to degrade more slowly, as established by the comparison of the difference in the values of $T_{x\%}$ for each different fiber concentration (Table 5). This fact could be due to a retardant action on the degradation of composites induced by the fibers. Natural fibers generally have low thermal conductivity and diffusivity [41] and could act as a good thermal barrier. In addition, the superstructure of lignin and cellulose fibrils in the composites could block the degradation products in the material volume as TGA proceeds. However, it is important to note that different thermal degradation mechanisms take place due to the mutual presence of PCL and fibers, and therefore a variation of the activation energy of the process could be expected [42]. The final residue measured at 700 °C increases coherently with the fiber percentage, also considering the final residue measured by TGA of raw fiber.

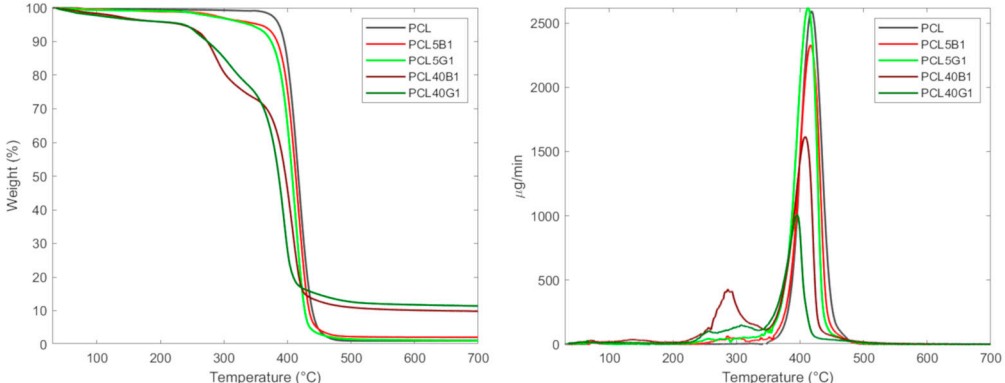

**Figure 11.** TGA thermograms (**left**) and the relative DTG (**right**) of PCL reference and composites.

**Table 5.** Comparison of TGA results of PCL reference and composites.

| Samples | $T_{onset}$ (°C) | $T_{5\%}$ (°C) | $T_{30\%}$ (°C) | $T_{50\%}$ (°C) | $T_{80\%}$ (°C) | Residue % | $T_{max}$ (°C) * | Other Peak (°C) |
|---------|---|---|---|---|---|---|---|---|
| | | | **From TGA** | | | | **From DTG** | |
| PCL ** | 395.3 | 386.5 | 408.1 | 417.1 | 431.2 | 1.1 | 418.0 | - |
| PCL5B1 | 264.2 | 355.6 | 403.4 | 413.0 | 426.8 | 2.1 | 417.1 | - |
| PCL40B1 | 263.6 | 233.9 | 367.3 | 397.0 | 418.0 | 9.9 | 410.0 | 287.5 |
| PCL5G1 | 239.2 | 340.9 | 397.2 | 407.3 | 420.9 | 1.2 | 411.6 | - |
| PCL40G1 | 239.2 | 237.2 | 362.8 | 386.1 | 409.2 | 11.4 | 395.1 | 256.0 308.0 |

* B1 fibers show $T_{max}$ at 286.4 °C and two shoulders at 261.2 and 325.7 °C; G1 fibers show $T_{max}$ at 310.1 °C and a peak at 252.8 °C (see Tables 2 and 3). ** The values found are in accordance with those reported by Ruseckaite et al. [40].

### 3.2.3. Mechanical Analyses

The four composites and the reference samples underwent tensile strength tests (Figure 12). The results obtained in terms of Young's modulus (E), elongation at break (%), and tensile strength (MPa) are reported in Table 6 and Figure 12. As far as PCL is concerned, the values obtained are in line with the values reported in the literature [30].

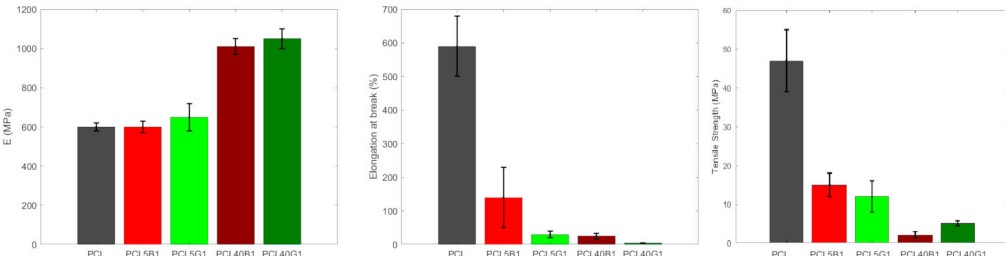

**Figure 12.** Young's modulus, elongation at break, and tensile strength of PCL reference and composites.

**Table 6.** Stress–strain results (Young's modulus E, elongation at break and tensile strength) of PCL reference and composites.

| Samples | Young's Modulus (MPa) | Elongation at Break (%) | Tensile Strength (MPa) |
|---|---|---|---|
| PCL | $600 \pm 20$ | $590 \pm 90$ | $47 \pm 8$ |
| PCL5B1 | $600 \pm 30$ | $140 \pm 90$ | $15 \pm 3$ |
| PCL40B1 | $1010 \pm 40$ | $25 \pm 8$ | $2.1 \pm 0.8$ |
| PCL5G1 | $650 \pm 70$ | $30 \pm 10$ | $12 \pm 4$ |
| PCL40G1 | $1050 \pm 50$ | $4.2 \pm 0.9$ | $5.1 \pm 0.6$ |

A one-way ANOVA test conducted on the different sample measured values allowed the evaluation of the statistical significance between possible different data groups of each sample with respect to that of PCL. The *p*-value corresponding to the F-statistic of one-way ANOVA is lower than 0.05 for each investigated property (Young's modulus, elongation at break, tensile strength), suggesting that one or more data group is significantly different from the others. The Tukey HSD test was subsequently carried out in order to identify which pairs of data groups were significantly different. The critical values of the Tukey HSD test were established at a significance level of $\alpha = 0.05$ by considering five different data groups, each consisting of 10 measurements and, therefore, 45 degrees of freedom. As for Young's modulus values, all composites result to be significatively different to PCL reference except that of PCL5B1, which results to be not significatively changed. The addition of 5% wt of both fibers had a slight effect on the final values measured; however, despite the low difference, PCL5G1 Young's modulus is significantly higher than that of PCL. The increase of Young's modulus is higher for composites with 40% of added fibers. Young's modulus values increase by about 400 MPa, as expected, due to the strengthening action of the fibers. The differences in the values between the PCL40B1 and PCL40G1 results are due to statistical fluctuations, so the addition of 40% wt, regardless of the type of fiber used, did not statistically affect Young's modulus. Elongation at the break of each sample results to be significantly different for PCL reference. In general, the decrease in the elongation at break as a result of the increasing stiffness of composites is apparent [43]. The addition of B1 fiber also, independently of their amount, caused a significant decrease in elongation while maintaining the modulus. Furthermore, G1 fibers induced a greater stiffness of the composites with respect to B1 fibers. This is consistent with the different composition of the fibers and their higher concentration, mainly containing cellulose (52%) in G1 and hemicellulose (68%) in B1. As for tensile strength, all the samples' results were significantly different from the PCL reference. Compared to PCL, adding fiber decreases tensile strength. However, the type of fiber does not influence the value, which decreases as the amount of fiber in the composite increases.

## 4. Conclusions

Fibers from processing byproducts of two different *Phaseolus vulgaris* varieties, named green and borlotti bean fibers, were used as reinforcing material for the preparation of Polycaprolactone (PCL)-based biocomposites by melt blending. Both fibers showed different characteristics of composition and morphology: borlotti bean fibers resulted rich

in hemicelluloses from chemical fractioning, with an amorphous component arranged in a spherical structure. Green bean fibers, on the other hand, are rich in cellulose, the crystalline component of the fiber, and show a fibrous structure. Two different composites were prepared for each fiber with 5% or 40% content calculated on the total mass of the sample. The different morphology of the fibers was maintained in the composites, even if the different compositions affected thermal and mechanical properties.

In detail, with the increase in the green bean fibers amount, the melt viscosity of the composites increased as Young's modulus, together with the decrease in the elongation at break. Tensile strength decreases by adding fibers. The type of fiber does not affect this value, which, however, decreases with the amount of fiber in the composite increases.

Likewise, the composites containing borlotti bean fibers showed a slightly higher percentage of crystallinity and greater thermal stability than the corresponding samples with green bean fibers.

In conclusion, this work highlights an increase in the viscosity in the melt, in the crystallinity, and in Young's modulus in the composites due to the presence of the fibers. At the same time, the thermal stability, elongation at break, and tensile strength undergo a decrease, and the characteristic temperatures of the composites remain aligned with those of the PCL.

In light of these results, a possible field of application of this type of materials can be envisaged in the field of rigid packaging or the agricultural field after the industrial scale-up of formulations.

**Author Contributions:** Conceptualization, L.R., C.D.M. and S.B.; methodology, L.R., C.D.M., L.A. and S.B.; software, L.A. and L.R.; validation, L.R., C.D.M. and S.B.; formal analysis, A.C., L.A., L.R. and C.D.M.; resources, S.B.; data curation, L.A. and L.R.; writing—original draft preparation, L.A.; writing—review and editing, L.A., L.R., C.D.M. and S.B.; funding acquisition, S.B. All authors have read and agreed to the published version of the manuscript.

**Funding:** This work was made possible with funding received from the European Union's Project LEGUVAL (Valorisation of legumes co-products and by-products for package application and energy production from biomass) under Capacities Project, Research for the benefit of SME Associations, FP7-SME-2012 (GA 315241).

**Data Availability Statement:** The data that support the findings of this study are available from the authors upon reasonable request.

**Acknowledgments:** The authors would like to thank Patrizia Cinelli (Department of Civil and Industrial Engineering University of Pisa) for having kindly donated the PCL for the preparation of the composites. The authors thank Luca Pardi (CNR-IPCF) for the revision of the English language of the manuscript.

**Conflicts of Interest:** The authors declare no conflict of interest. The funders had no role in the design of this study; in the collection, analyses, or interpretation of data; in the writing of the manuscript, or in the decision to publish the results.

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
