# Peer review of "Agro-Waste Bean Fibers as Reinforce Materials for Polycaprolactone Composites"

_compounds, doi:10.3390/compounds3030036_

Round 1

Reviewer 1 Report

I reviewed the manuscript entitled “Agro-waste bean fibers as reinforce material for Polycaprolactone composites”.

This article studies the preparation of Polycaprolactone (PCL)-based bio-composites by melt blending using green bean and Borlotti bean fibers as reinforcing material.

This is a clear and well-written manuscript. On the other hand, the manuscript needs some modifications before accepting for publication.

1. The Title of the study reflects the topic as a whole and is sufficient.

2. The abstract part of the paper should be revised.

3. The manuscript does require some pruning and a thorough editing for proper grammar, punctuation and spelling. Thus, the introduction part should be thoroughly checked for spelling and sentence mistakes.

4. Figures 3, 4, 5, and 6 are not clear. I request that you redraw them.

5. Figures 7,10,11 and 12: It would be good to compare the values measured here with corresponding literature values.

6. The References part is adequate, but there is a very little reference to 2020, 2021 and 2022, these interesting recent papers should be cited:

ü  Khmais Zdiri, Adel Elamri, Omar Harzallah, Mohamed Hamdaoui & Jocelyn Brendlé (2021) Properties of recycled PP/clay filaments used for simulation of wastewater treatment filter, The Journal of The Textile Institute, 112:11, 1753-1762, DOI: 10.1080/00405000.2020.1841946

ü  Zdiri, K., Elamri, A., Harzallah, O., Hamdaoui, M. (2022). Thermo-Mechanical Characterization of Post-consumer PP/Tunisian Organo-Clay Filaments. In: Msahli, S., Debbabi, F. (eds) Advances in Applied Research on Textile and Materials - IX. CIRATM 2020. Springer Proceedings in Materials, vol 17. Springer, Cham. https://doi.org/10.1007/978-3-031-08842-1_7

ü  Zdiri, K.; Elamri, A.; Hamdaoui, M.; Harzallah, O.; Khenoussi, N.; Brendlé, J. Valorization of Post-consumer PP by (Un)modified Tunisian Clay Nanoparticles Incorporation. Waste Biomass Valor 11, 2285–2296 (2020). https://doi.org/10.1007/s12649-018-0427-2

ü  Zdiri, K.; Cayla, A.; Elamri, A.; Erard, A.; Salaun, F. Alginate-Based Bio-Composites and Their Potential Applications. J. Funct. Biomater. 2022, 13, 117. https://doi.org/10.3390/jfb13030117

The manuscript does require some pruning and a thorough editing for proper grammar, punctuation and spelling. Thus, the introduction part should be thoroughly checked for spelling and sentence mistakes.

Reviewer 2 Report

The work is interesting in terms of knowledge. Particularly interesting is the broad characteristics of the tested types of bean fibers.

However, I have a few comments, questions and suggestions for completing the work.

„Introduction”- The authors did not include any information on whether composites containing bean fibers, e.g. with other materials, had been tested before. Is the pcl/bean fiber composition completely new?

„Materials” - listing only the PCL manufacturer is not enough. What type of PCL is it? What are its basic properties and uses provided by the manufacturer?

”PCL composite preparation”- What was the humidity of the ingredients after drying? Processing methods used in laboratories and not in industry (such as extrusion, injection molding ) were used. Nothing is known about the cooling of samples after pressing. The cooling process is affected by on the structure, the share of the crystalline phase. Thus, it has an impact on the properties of the samples obtained.

The scope of the research does not include the basic MFR processability index. Thus, it is not possible to assess whether the tested compositions can be industrially processed by extrusion or injection molding.

Providing mechanical properties only in terms of Young's modulus and elongation at break significantly limits information about the properties of the composition, and thus about potential applications. Why is the basic result of stretching, which is tensile strength, not included? Also interesting and expected properties are hardness, impact strength, HDT or VST temperature.

Taking into account the obtained test results "TGA/DTG profiles of raw G1 and B1 fibers indicated their processability up to 200°C". Therefore, a limitation in the application of the composition (limited use temperature) is the PCL matrix material with a melting temperature of 57.4 °C. So, was the use of PCL as a matrix a good choice? After all, there are other biodegradable plastics with higher heat resistance and at the same time processed in a range that is safe for the tested fillers.

„Conclusions” - what potential applications do the authors anticipate for the tested compositions?
